

# Climatic responses to systematic time variations of parameters: A dynamical approach

Catherine Nicolis[1]

[1]Institut Royal Météorologique de Belgique, 3 av. Circulaire, 1180 Brussels, Belgium

**Correspondence:** Catherine Nicolis (cnicolis@oma.be)

**Abstract.** The climatic response to time-dependent parameters is revisited from a nonlinear dynamics perspective. Some general trends are identified, based on a generalised stability criterion extending classical stability analysis to account for the presence of time-varying coefficients in the evolution equations of the system's variables. Theoretical predictions are validated by the results of numerical integration of the evolution equations of prototypical systems of relevance in atmospheric and climatic dynamics.

## 1 Introduction

The climatic impact of systematic variations of certain key parameters in time arising from anthropogenic effects such as increasing $CO_2$ concentration constitutes currently a major scientific, economic and societal issue (Goodie and Guff, 2001). There exists a vast literature on the subject culminating in the derivation of a number of scenarios of future climatic change, based on the integration of detailed numerical models and on the intercomparison of their respective predictions (Andrews et al, 2012).

On the other hand, it is widely recognized that the atmosphere and climate are highly nonlinear systems subjected to intricate feedbacks giving rise to a rich variety of complex dynamical behaviors such as self-generated periodicities, deterministic chaos, or transitions between different states (Nicolis and Nicolis, 1987; Dijkstra, 2013). A major advance of nonlinear dynamics has been to show that these behaviors often rest on a limited number of generic, global features independent of details concerning individual processes (Guckenheimer and Holmes, 1983). This suggests that it might be of interest to search for regularities likely to recur across different models and scenarios that could possibly be masked in a detailed full-scale analysis. In this work we revisit the climatic response to time-dependent parameters from such a nonlinear dynamics perspective, extending an early investigation in this direction by the present author (Nicolis, 1988).

The starting point is a set of equations governing the evolution of the atmospheric and climatic variables. We consider a reference state corresponding to a solution of these equations for some particular values of the parameters. We next switch on a systematic variation of these parameters in time and follow the subsequent transient response of the reference state to this forcing. The questions we raise are, whether and if so for how long the system will follow passively this variation while remaining in the same branch of states; under what conditions it will jump to a new regime and if so when this transition will



occur; and finally, whether states that would otherwise prevail in the absence of parameter variation are altered significantly or missed altogether.

A general formulation for addressing these questions is outlined in Section 2, where a generalized stability criterion for remaining or not in the vicinity of the reference state is derived and some general scenarios of subsequent evolution are

discussed. In the light of these ideas the response to time varying parameters is analyzed in Sections 3 to 5 in situations giving rise to oscillatory behavior, to chaotic behavior and to transitions between simultaneously stable states. The main conclusions are summarized in Section 6.

Throughout her career Anna Trevisan has managed to combine harmoniously theoretical ideas and tools and large-scale numerical approaches to tackle fundamental problems of concern in atmospheric physics. This paper is dedicated to her memory.

## 2  Formulation

Let $\{x_i\}$, $i = 1, \cdots, n$ be the set of atmospheric/climatic variables and $\lambda_\alpha$, $\alpha = 1, \cdots, r$ a set of parameters characteristic of the rates of the various processes involved in the evolution of these variables. The rate of change of $\{x_i\}$ in time will be given by a set of equations of the form

$$\frac{dx_i}{dt} = F_i(\{\lambda_\alpha\}, \{x_j\}) \qquad i, j = 1, \cdots, n \tag{1}$$

where the evolution laws $\{F_i\}$ are, typically, nonlinear functions of the $\{x_j\}$.

We are interested in situations in which one of these parameters varies systematically in time as a result of an externally-induced forcing of natural or anthropogenic origin. The particular form of variation we shall focus on is a slow variation in the form of a *ramp*,

$$\lambda(t) = \lambda_0 + \epsilon t \qquad \epsilon << 1 \tag{2}$$

where $t$ is the time and $\lambda_0$ the value of $\lambda$ prevailing at a stage where the evolution of the $\{x_i\}$ is started. Introducing the slow time scale

$$\tau = \epsilon t \tag{3a}$$

one may cast Eqs. (1) in the form

$$\epsilon \frac{dx_i}{d\tau} = F_i(\lambda(\tau), \{x_j\}) \tag{3b}$$

where from now on we will discard all parameters other than the time-varying one $\lambda(\tau)$.



Following the procedure outlined in the Introduction we consider now a particular, possibly time-dependent state $\{\overline{x}_i\}$ lying at $t = 0$ on an invariant attracting set of states corresponding to the value $\lambda_0$ of parameter $\lambda$ and switch on next the change of $\lambda$ in time according to Eq. (2). We are interested in the response of the reference state $\{\overline{x}_i\}$ to this change as defined by the instantaneous deviations from it $\{\delta x_i\}$, initially assumed to be small. Writing

$$x_i = \overline{x}_i + \delta x_i \tag{4a}$$

and substituting into Eq. (3b) one obtains then a linearized set of equations of the form

$$\epsilon \frac{d\delta x_i}{d\tau} = \sum_j J_{ij}(\tau)\delta x_j \tag{4b}$$

where $J_{ij} = (\partial F_i/\partial x_j)_{\{\overline{x}_j\}}$ are the elements of the Jacobian matrix associated to $F_i$. These quantities depend on $\tau$ through the time dependence of $\lambda$ (Eq. (2)) and possibly through the fact that the reference state $\{\overline{x}_i\}$ may itself be part of a periodic or
chaotic attractor. In what follows we will be especially interested in the dependence on $\tau$ induced by $\lambda$.

Equations (4b) constitute a set of coupled equations with slowly varying coefficients. Generalizing the time-exponential solutions familiar from classical stability analysis we seek for solutions of these equations of the WKB form (Kevorkian and Cole, 1996)

$$\delta x_i(\tau) = \exp[\frac{1}{\epsilon}\Phi(\tau)]A_i(\epsilon,\tau) \tag{5}$$

where the amplitudes $A_i$ depend smoothly on $\epsilon$. Substituting into (4b) we obtain, to the dominant order in $\epsilon$,

$$A_i \frac{d\Phi}{d\tau} = \sum_j J_{ij}(\tau)A_j$$

It follows that the quantity

$$w(\tau) = \frac{d\Phi}{d\tau} \tag{6a}$$

satisfies the generalized characteristic equation

$$\det|J_{ij}(\tau) - w(\tau)\delta_{ij}^{\mathrm{kr}}| = 0 \tag{6b}$$

and plays thus the role of a generalized eigenvalue of the (time-dependent) Jacobian matrix $J(\tau)$.


We are now in the position to derive the condition under which the response $\{\delta x_i(\tau)\}$ will remain bounded or will, on the contrary, show explosive behavior. Taking Eq. (5) into account one sees straightforwardly that the threshold separating these two regimes is given by the relation

$$\mathrm{Re}\Phi(\tau_c) = \int_0^{\tau_c} d\tau' \mathrm{Re}w(\tau') = 0 \tag{7}$$

This relation, if satisfied, defines a critical time $t_c = \epsilon\tau_c$ and a corresponding critical value $\lambda_c = \lambda_0 + \epsilon t_c$ of parameter $\lambda$ beyond which the system will depart from the reference state and evolve toward a new branch of solutions. We expect that these solutions will be part of the bifurcation diagram of the dynamical system defined by Eqs. (1). The question will then be, how these solutions are reached if one moves across this bifurcation diagram according to Eq. (2), starting from a stable branch of solutions. In particular, are the transitions toward the new states taking place in the "static" bifurcation points of Eqs. (1);

and if not, in the "dynamical" view of bifurcation adopted here, are the transitions advanced, delayed or skipped altogether (Erneux and Mandel, 1986; Baer et al., 1989; Benoit, 1991; Nicolis and Nicolis, 2004, 2014). Failure to satisfy relation (7) for any $\tau$ within a certain range, starting at $\tau = 0$ from a stable branch of solution would on the other hand imply that the system will remain on this branch of solutions for this time period. One would then like to know how is the structure of this solution affected as the parameter $\lambda$ is varying in time. In particular, can this time dependence lead eventually to catastrophic behavior,

by e.g. enabling the system to cross threshold values that would otherwise never be reached.

In what follows these questions will be addressed for selected classes of systems giving rise to periodic behavior, to chaotic dynamics and to transitions between simultaneously stable steady states. We stress that the logic underlying our formulation differs from the one adopted in typical general circulation model-based experiments (Gregory et al., 2015) in which, e.g., $C0_2$ concentration is suddenly increased (CMIP5 abrupt $n \times C0_2$ experiments where $n$ is typically 2 or 4) and the system is

subsequently left to relax to its final state keeping this concentration constant.

## 3   Periodic behavior

A dynamical system giving rise to sustained oscillations must involve at least two coupled variables. The onset of oscillatory behavior will occur through a Hopf bifurcation, in the vicinity of which the Jacobian matrix associated to the rate functions $\{F_i\}$ in Eqs. (1) possesses two complex conjugate eigenvalues whose real parts become positive beyond the bifurcation point

(Guckenheimer and Holmes, 1983). An interesting example of Eqs. (1) of relevance in climate theory giving rise to this type of behavior is the sea ice-ocean surface temperature model developed by Saltzman, Sutera and Hansen (Saltzman et al., 1982) which in appropriate rescaled variables reads (Nicolis, 1984)

$$\begin{aligned}
\frac{d\eta}{dt} &= -\eta + \theta \\
\frac{d\theta}{dt} &= -a\eta + b\theta - \eta^2\theta
\end{aligned} \tag{8}$$



Here $\eta$ represents the deviation of the sine of the latitude of sea ice extent from the reference steady state and $\theta$ the excess mean ocean surface temperature. $a$, $b$ are positive parameters describing, respectively, the negative feedback of ice extent on temperature and the positive feedback of temperature on itself. Finally, $\eta^2\theta$ accounts for nonlinear restoring mechanisms.

Previous studies have shown that as long as $a > b$ the steady-sate solution $\eta = \theta = 0$ of Eqs. (8) is stable for values of the parameter $b$ less than 1 and loses its stability through a Hopf bifurcation toward time-periodic solutions at a critical value $b_c^{(0)} = 1$.

In the context of the present work it will be natural to choose $b$ as the time-dependent parameter

$$b = b_0 + \epsilon t \tag{9}$$

We choose again as reference state the steady-state solution $\eta = \theta = 0$ and a starting value $b_0$ for which this state is stable
($b_0 < 1$) and seek for solutions of Eqs. (8) when the time dependence of $b$ is switched on according to Eq. (9) in the WKB forms of Eq. (5). One obtains then straightforwardly the following explicit form of the generalized characteristic equation (Eq. (6b))

$$\left(\frac{d\Phi}{d\tau}\right)^2 - (b_0 + \tau - 1)\frac{d\Phi}{d\tau} + (a - (b_0 + \tau)) = 0 \tag{10}$$

where we have again set $\tau = \epsilon t$. In view of our choice $b_0 < 1$ and $a > b$ there exists a range of values of $\tau$ for which this
equation admits complex conjugate roots $(d\Phi/d\tau)_\pm$. The stability criterion expressed by Eq. (7) in terms of the real part of these solutions leads then to the explicit form

$$
\begin{aligned}
\mathrm{Re}\Phi_\pm(\tau_c) &= \int_0^{\tau_c} d\tau' \mathrm{Re}\left(\frac{d\Phi}{d\tau'}\right) \\
&= \frac{1}{2}\int_0^{\tau_c} d\tau'(b_0 - 1 + \tau') \\
&= \frac{1}{2}\left[(b_0 - 1)\tau_c + \frac{\tau_c^2}{2}\right] = 0
\end{aligned}
\tag{11}
$$

This relation determines a critical time

$$t_c = \frac{2(1 - b_0)}{\epsilon} \tag{12a}$$

and a new critical parameter value

$$b_c = b_0 + \epsilon t_c = 2 - b_0 \tag{12b}$$





independent of $\epsilon$, beyond which the system will leave the reference state and evolve toward a periodic solution. The point is that (a), unless $b_0 = 1$, $b_c$ is different from the value $b_c^{(0)} = 1$ corresponding to the "static" Hopf bifurcation point; and (b), as a result the transition to the instability region is postponed for a time interval proportional to the distance of $b_c^{(0)}$ from the starting value $b_0$ and inversely proportional to the smallness parameter $\epsilon$. During this delay the system will keep following the initial

branch of states, which in the classical setting of time-independent parameter $b$ would be unstable and is now temporarily stabilized. In a climate dynamics perspective one could rephrase this result by the statement that rather than precipitating the system to the instability that was bound to occur at $b_c^{(0)} = 1$ and to the large deviations in the form of oscillations that would follow, the time-dependent forcing has on the contrary postponed this "catastrophe". Everything happens as if the presence of the time-dependent forcing during the time spent in the stable region enhances the "inertia" of the system and hence its further

stabilisation into this region. This realization illustrates how long term predictions can interfere in a subtle and unexpected manner with the dynamical complexity of the underlying system.

We now confront these predictions to the results of direct numerical integration of Eqs. (8) with parameter $b$ varying according to Eq. (9). Figure 1 depicts the evolution of variable $\eta$ in a representation where time enters through the parameter $b = b_0 + \epsilon t$, with $a = 4$, $b_0 = 0$ and $\epsilon = 0.01$. As can be seen the system follows the state $\eta = 0$, runs across the static Hopf

bifurcation point $b_c^{(0)} = 1$ as if nothing was happening and finally jumps to an oscillatory state at a time corresponding to $b = 2$, in full agreement with the theoretical result of Eqs. (12a)-(12b). On the other hand, when the transition is finally taking place the system is rapidly precipitated in a regime of large amplitude oscillations, much larger than those that would start smoothly at $b_c^{(0)} = 1$ in the classical setting of a static Hopf bifurcation. We witness, in some sense, a payoff between the postponement and the extent of a potentially catastrophic event.

These results hold for a wide range of values of $\epsilon$, but at some point one witnesses deviations from the asymptotic regime as captured by the WKB type of solutions. The trend, as illustrated in Fig. 2, is that for increasing $\epsilon$ (here $\epsilon = 0.1$) the transition to oscillations is further postponed beyond the value predicted by our theoretical estimate. We conjecture that this is due to the fact that the bifurcation diagram is now traversed faster than the characteristic growth rates of perturbations that would otherwise remove the system from the reference state. These perturbations are thus temporarily quenched until their growth

rate becomes substantial and can no longer be counteracted by moving across the bifurcation diagram.

A question related to the foregoing and of interest in the context of atmospheric and climate dynamics is, when a particular variable of relevance in a system subjected to a systematic time-dependent forcing will cross for the first time a certain prescribed level. Figure 3 summarizes the results obtained by numerically integrating Eqs. (8)-(9) for a wide range of values of the ramp parameter $\epsilon$ and for a threshold value $|\eta| = 1$ set for the variable $\eta$ of the model. We observe an ascending trend with

increasing $\epsilon$ values, which can be explained qualitatively by the arguments advanced in connection with Fig. 2.

## 4 Chaotic dynamics

Chaotic dynamics is ubiquitous in the atmosphere, where it is responsible for the growth of prediction errors arising from small uncertainties in the initial conditions (Lorenz, 1984). There are strong arguments supporting the view that it also underlies a



host of large scale phenomena responsible for climatic variability (Tsonis, 1992; Essex and McKitrick, 2007). In the present section we analyze the effect of a systematic time variation of parameters on a simplified model of thermal convection giving rise to chaotic behavior due to Lorenz (Lorenz, 1963) in which the velocity and temperature fields are expanded in Fourier series keeping one Fourier mode for the vertical component of the velocity (variable $x$, see below) and two Fourier modes for

the temperature variation (variables $y$ and $z$). One arrives then at the equations

$$
\begin{aligned}
\frac{dx}{dt} &= \sigma(-x+y) \\
\frac{dy}{dt} &= rx - y - xz \\
\frac{dz}{dt} &= xy - bz
\end{aligned}
\tag{13}
$$

The parameters $\sigma$ and $r$ are scaled Prandtl and Rayleigh numbers, respectively, and $b$ accounts for the geometry of the convec-

tive pattern.

Equations (13) have been studied extensively in the literature(Sparrow, 1982). We briefly summarize some results that will be relevant for our purposes.

(i) The steady state $x = y = z = 0$ (where convection is absent) is stable for $r < 1$ and loses its stability at $r = 1$ through a

pitchfork bifurcation.

(ii) Beyond $r = 1$ a pair of non-trivial steady states representative of convection emerges, given by $x_\pm = y_\pm = \pm\sqrt{b(r-1)}$, $z = r - 1$. These states remain stable for $r$ less than a threshold value $r_T^{(0)} = \sigma(\sigma + b + 3))/(\sigma - b - 1)$.

(iii) At $r = r_T^{(0)}$ a Hopf bifurcation is occurring, but the branches of periodic solutions are subcritical (i.e., exist for $r < r_T^{(0)}$) and thus unstable.

(iv) Beyond $r_T^{(0)}$ one observes a variety of complex chaotic behaviors which emerge suddenly as global, finite amplitude solutions.


In what follows it will be natural to consider $r$, which incorporates the effect of the thermal constraints acting on the system, as time-dependent parameter.

Setting

$r = r_0 + \epsilon t$     (14)





we choose as reference state one of the convective states, say $(x_-, y_-, z)$, and a starting value $r_0$ for which this state is stable, i.e., $1 < r_0 < r_T^{(0)}$. Similarly to Section 3 we seek for solutions of Eqs. (13) with time-dependent $r$ according to Eq. (14) in the WKB form of Eq. (5). We obtain in this way the following explicit form of the generalized characteristic equation (6) associated to the Jacobian matrix of Eqs. (13) around $(x_-, y_-, z)$ :

$$
(\frac{d\Phi}{d\tau})^3 + (\sigma + b + 1)(\frac{d\Phi}{d\tau})^2
$$
$$
+ b(\sigma + r_0 + \tau)(\frac{d\Phi}{d\tau}) + 2b\sigma(r_0 + \tau - 1) = 0 \tag{15}
$$

where $\tau = \epsilon t$. The system will leave the reference state at a critical time $t_c = \tau_c/\epsilon$ and a critical, $\epsilon$-independent parameter value $r_c = r_0 + \epsilon t_c$ determined by relation (7),

$$
\mathrm{Re}\Phi(\tau_c) = \int_0^{\tau_c} d\tau' \mathrm{Re}(\frac{d\Phi}{d\tau'}) = 0 \tag{16}
$$

where $d\Phi/d\tau$ as a function of $\tau$ is given by Eq. (15).

Figure 4 summarizes the results obtained by numerical evaluation of the integral in Eq. (16). We have set for this purpose $\sigma = 10$, $b = 8/3$ in Eq. (15). The static Hopf bifurcation point $r_T^{(0)}$ corresponding to these values is $r_T^{(0)} \approx 24.74$. We choose $r_0$ values in the interval (10, 24) prior to this value, for which Eq. (15) in the absence of time-dependent parameter possesses a real negative root and a pair of complex conjugate roots with negative real part. We then plot in the figure the critical value of

$r$ of the onset of chaotic solutions, $r_c = r_0 + \tau_c$, as a function of $r_0$. As can be seen $r_c$ decreases quasi-linearly with $r_0$, from a value of about 45 at $r_0 = 10$ to a value of about 25.5 at $r_0 = 24$.

Figure 5, to be compared with Fig. 1, depicts the evolution of variable $x$ versus time (expressed in terms of parameter $r = r_0 + \epsilon t$) as obtained from direct numerical integration of Eqs. (13)-(14) for $r_0 = 20$ and $\epsilon = 0.01$. Once again the system runs across the static transition point $r_T^{(0)} \approx 24.74$, remains close to the reference state $(x_-, y_-, z)$ and eventually evolves

toward a chaotic state at a time corresponding to a value $r_c$ between 29 and 30, in excellent agreement with the theoretical predictions summarized in Fig. 4. This result remains robust in the sense that $r_c$ is essentially determined by $r_0$ independent of $\epsilon$ for a wide range of $\epsilon$ values. But as $\epsilon$ is increased one witnesses deviations from the theoretical estimate as illustrated in Fig. 6, where for the same value of $r_0$ as before and for $\epsilon = 0.1$ the transition to the chaotic regime occurs at a value of $r$ of about 32.

Related to the foregoing is the question, when the variable $x$ will cross for the first time a certain prescribed level higher than its value in the reference state. Figure 7 summarizes the results obtained by numerically integrating Eqs. (13)-(14) for a wide range of values of $\epsilon$ and for a threshold value of $|x/x_\pm| = 1.5$. We observe an increasing trend similar to the one reported in Fig. 3, reflecting the enhancement of stabilization of the reference state upon increasing the rate at which the bifurcation diagram is transversed.

Assuming now that the system has settled in the chaotic regime, we wish to quantify in some way the effect of the time variation of parameter $r$ on the behavior of the principal variables involved. A first result in this direction is reported in Fig. 8a,




where the instantaneous ensemble averages over 100,000 initial conditions lying on the initial attractor of $x$, $y$ and $z$ are plotted against time as measured again by $r = r_0 + \epsilon t$ for values between 26 and 36, for which the system shows chaotic behavior. We see that $x$ and $y$ hardly perceive the time-dependent forcing, whereas $z$ follows it in a rather straightforward manner. This shows how subtle the response of system to a parameter may be. Notice, however, that a further increase of $r$ may bring the

system to a new attractor and change the qualitative features of the dynamics.

Figure 8b depicts the time evolution (again via the dependence on $r(t)$) of the variances of $x$, $y$ and $z$ variables around their means. We see that they all follow a systematic increasing trend. This suggests the possibility that variance can serve as a key quantity and as an early warning of future changes induced by a time-dependent forcing, especially as far as the occurrence of extreme events is concerned (Chavez et al., 2016).

## 5  Transitions between states and limit point bifurcations

There is ample evidence of large-scale climatic transitions between glacial and interglacial regimes (Berger, 1981). On a shorter time scale transitions between different global circulation patterns associated to the phenomenon of persistent flow regimes at mid-latitudes, also referred as "blocking" in contrast to the familiar zonal flows, are well documented and constitute one of the principal elements of low frequency atmospheric variability.

In this section we analyze the effect of systematic time variations of parameters in the classic three-variable model of the zonal to blocking transitions that goes back to the pioneering work of Charney (Charney and De Vore, 1979). The model consists in expanding the stream function $\psi$ associated to the horizontal velocity field in series of orthogonal functions and, upon substituting into the equation for the potential vorticity, truncating the resulting infinite system of equations for the coupled modes to the first three ones. One obtains in this way a system of equations of the form

$$\frac{d\psi_A}{dt} = -k(\psi_A - \psi_A^*) + h_1\psi_L$$
$$\frac{d\psi_K}{dt} = -(\alpha\psi_A - \beta)\psi_L - k\psi_K$$
$$\frac{d\psi_L}{dt} = (\alpha\psi_A - \beta)\psi_K - h_2\psi_A - k\psi_L \qquad (17)$$

Here $\psi_A$, $\psi_K$, $\psi_L$ denote the amplitudes of the three retained modes, $\psi_A^*$ is a forcing parameter of the flow and $k$ accounts for the effect of the dissipation. The remaining parameters are related to the topography and to the mean height of the fluid layer.

Higher order truncation schemes have been developed by Ghil and coworkers (Legras and Ghil, 1985).

Figure 9 depicts the bifurcation diagram of model (17) in which the zonally averaged velocity mode $\psi_A$ is plotted against the forcing parameter $\psi_A^*$, keeping the other parameters fixed (see caption). One observes two branches of stable solutions (full lines) colliding and terminating with an intermediate unstable branch (dashed line) at two critical values corresponding to a limit point bifurcation. Going back to the space dependence of the velocity field one finds that the lower branch corresponds to

the state of atmospheric blocking whereas the upper branch is representative of zonal flow (Charney and De Vore, 1979; Egger, 1981; Nicolis, 2002).



In what follows we choose $\psi_A^*$ as forcing parameter, setting

$$\psi_A^* = \psi_0^* + \epsilon t \qquad (18)$$

Figure 10 summarizes the results of numerical simulations of the full equations (17)-(18) for three different initial conditions that in absence of time variation of $\psi_A^*$ would all be attracted by the lower (stable) branch of solutions. We see that in actual

fact this branch is skipped altogether and the trajectories evolve to the upper stable branch passing through the intermediate unstable one. Interestingly, they are all significantly delayed before reaching eventually the upper branch. Part of this delay can be attributed to the slowing down of the dynamics in the vicinity of the limit point, where the generalized eigenvalues of the Jacobian around the upper branch tend to zero for $\psi_A^*$ tending to its value at the limit point.

A second series of numerical simulations is reported in Fig. 11, starting this time from a state in the vicinity of the lower

stable branch. For very small $\epsilon$ we see that the branch is followed up to the rightmost point whereupon the trajectory jumps to the upper branch with practically no delay. But as $\epsilon$ is increased one witnesses increasingly early departures from the reference state. The corresponding trajectories pass through the intermediate unstable branch and tend to the upper one reaching it with delays that increase markedly with $\epsilon$. This behavior reflects undoubtedly the weak stability properties of the lower branch which comes increasingly closer to the intermediate unstable one for increasing $\psi_A^*$ values. Furthermore, perturbations around

the lower branch undergo damped oscillations. Because of this the trajectory, entrained to a higher $\psi_A^*$ value under the effect of the ramp, may temporarily pass a threshold beyond which it starts being attracted by the upper branch.

A more quantitative explanation, albeit limited to the vicinity of the limit points, appeals to the fundamental result that in the vicinity of a limit point bifurcation the dynamics simplifies considerably. Specifically, there exists a single variable $z$ related to combinations of the three original variables appearing in Eqs. (17), to which one refers as order parameter, satisfying a

universal equation of the form (Guckenheimer and Holmes, 1983)

$$\frac{dz}{dt} = \mu(t) - z^2 \qquad (19)$$

where $\mu$ is a combination of $\psi_A^*$ and of the other parameters appearing in Eqs. (17)-(18).

Setting again $\mu(t) = \mu_0 + \epsilon t$ one can show that upon appropriate scaling of variables and parameters Eq. (19) can be transformed to an Airy equation (Davies and Krishna, 1996). The solution in terms of the original variable $z$ is then

$$z = \epsilon^{1/3} \frac{Ai'(\frac{\mu_0 + \epsilon t}{\epsilon^{2/3}}) + CBi'(\frac{\mu_0 + \epsilon t}{\epsilon^{2/3}})}{Ai(\frac{\mu_0 + \epsilon t}{\epsilon^{2/3}}) + CBi(\frac{\mu_0 + \epsilon t}{\epsilon^{2/3}})} \qquad (20)$$

Here $Ai$, $Bi$ are the Airy functions, the prime denotes derivative with respect to the whole argument and $C$ is determined by the initial condition $z(0) = z_0$. Carrying out at the level of Eq. (19) the numerical experiments summarized in Fig. 10 one can now delimit the initial conditions that will evolve to the upper stable branch $z = \sqrt{\mu(t)}$ of the quasi-static solution of Eq. (19), by requiring that the denominator in Eq. (20) remains different from zero, which in turn requires that $C$ be positive. This



yields trajectories behaving for the original dynamical system according to Fig. 10 (Nicolis and Nicolis, 2014). Notice that the approach outlined in Section 2 and applied successfully in Sections. 3 and 4 is not appropriate in the presence of a limit point, since the reference stable state does not continue beyond the bifurcation point as an unstable branch of solutions but disappears altogether.

## 6   Conclusions

In this work we identified some universal trends underlying the response of a system to systematic changes of parameters in time. Most prominent among them are that, starting with a stable branch of states, transitions to new regimes that would occur in the "static" case of absence of time variation of parameters tend to be delayed; states that in the static case are unstable are temporarily stabilized; and states that in the static case are stable can be skipped altogether. As a corollary, the times at which threshold values are first crossed have been obtained as a function of the rate of increase of the parameters in time.

These conclusions were based on a generalized stability criterion extending classical stability analysis to account for the presence of time-varying coefficients in the evolution equations of the system's variables, as well as on analytic solutions prevailing in the vicinity of transition points. They were validated by the results of numerical integration of the evolution equations of prototypical systems of relevance in atmospheric and climate dynamics giving rise to periodic behavior, to chaotic dynamics and to transitions between simultaneously stable steady states.

Throughout our approach the time variation of the parameters has been fully and consistently incorporated into the intrinsic time evolution of the system's variables as given by the appropriate rate equations. Our results depend critically on this view of parameter-system co-evolution, a scenario reflecting, we believe, the way a natural system is actually evolving in time. This scenario differs from those adopted in current studies on climatic change based on the integration of large numerical models, where parameters are suddenly set at a different level and the system is subsequently left to relax under these new conditions. It would be interesting to allow for different scenarios beyond the standard ones, closer to our fully dynamical approach, and to test the robustness of the conclusions reached under these different conditions.

*Competing interests.*   No competing interests are present

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



**Figure captions**

Fig. 1 Evolution of variable $\eta$ versus the instantaneous value of the feedback parameter $b$ as obtained numerically from model (8) in the presence of a time dependence $b = b_0 + \epsilon t$ with $b_0 = 0$, $a = 4$ and $\epsilon = 0.01$.

Fig.2 As in Fig.1 but $\epsilon = 0.1$.

Fig. 3 Instantaneous value of parameter $b$ corresponding to the first passage of variable $\eta$ from threshold $|\eta| = 1$ versus the intensity $\epsilon$ of the ramp parameter with $b_0 = 0$. Other parameter values as in Fig. 1 .

Fig. 4 Theoretical estimate of the onset of chaotic solutions versus the initial value of parameter $r_0$ (Eq. (15)) for model (13) in the presence of a time dependence in the form of $r = r_0 + \epsilon t$ with $\epsilon = 0.01$. Other parameter values $\sigma = 10$ and $b = 8/3$.

Fig. 5 Time evolution of variable $x$ of model (13) expressed in terms of the instantaneous value of $r$ with $\epsilon = 0.01$ and initial condition $x = x_-$, $y = y_-$, $z = r_0 - 1$.

Fig. 6 As in Fig. 5 but with $\epsilon = 0.1$.

Fig. 7 As in Fig. 3 but for model (13) with $r_0 = 20$ and threshold value $|x/x_\pm| = 1.5$. Other parameter values as in Fig. 4.

Fig. 8 Ensemble averages (a) and variances (b) of variables $x$, $y$, and $z$ of model (13) in the chaotic regime versus the instantaneous value of the ramp parameter $r$ starting from $r_0 = 26$ with $\epsilon = 0.01$. Number of initial conditions is 100,000.

Fig. 9 Bifurcation diagram of model (17) as parameter $\psi_A^*$ increases from 0.05 to 7. Full lines represent the two stable solutions, blocked (lower), zonal (upper) and dashed line the intermediate unstable state. Parameter values $k = 10^{-2}$, $\beta = 0.1$, $h_1 = 1.6\sqrt{2}/(3\pi)$, $h2 = h1/5$ and $\alpha = 8h1$.

Fig. 10 Time evolution of variable $\psi_A$ of model (17) in the presence of a time dependent forcing (Eq. (18)). Initial conditions (a),(b) and (c) evolve to the zonal state (upper stable branch of the bifurcation diagram) although in the absence of the time dependent forcing the system is bound to follow the blocked circulation solution (low stable branch of the bifurcation diagram). Parameter values $\epsilon = 0.01$ and as in Fig. 9.

Fig. 11 As in Fig. 10 but for initial conditions in the vicinity of the lower stable branch of the bifurcation diagram and three different $\epsilon$ values.

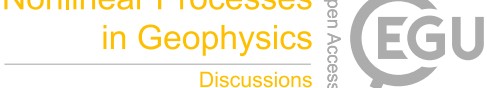



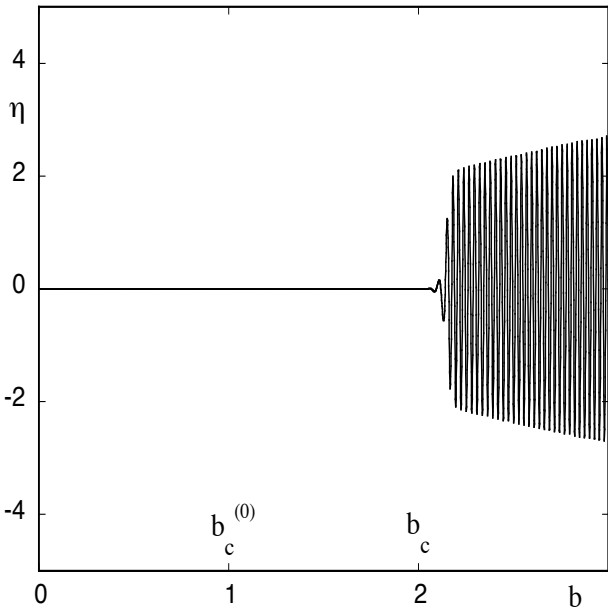

**Figure 1.**

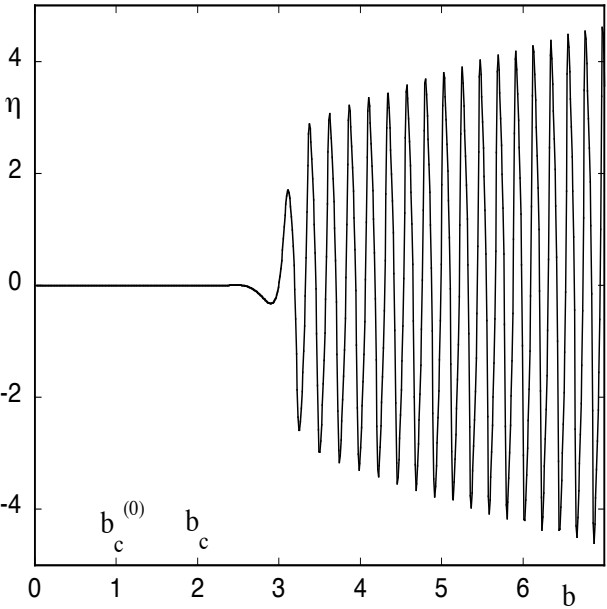

**Figure 2.**

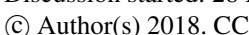

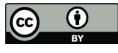

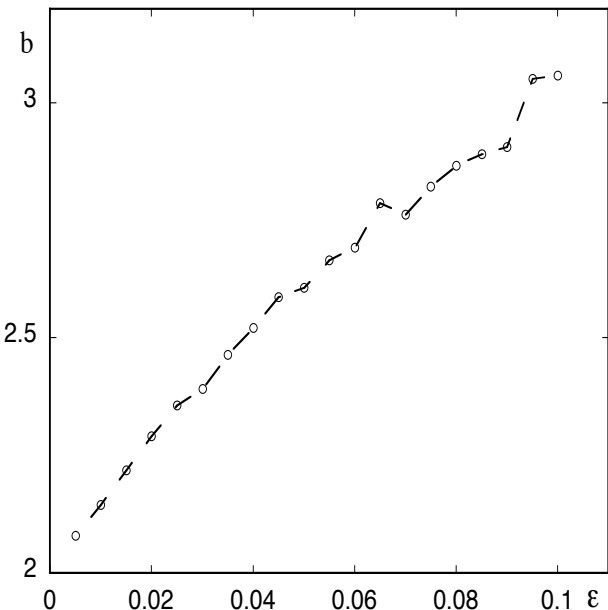

**Figure 3.**

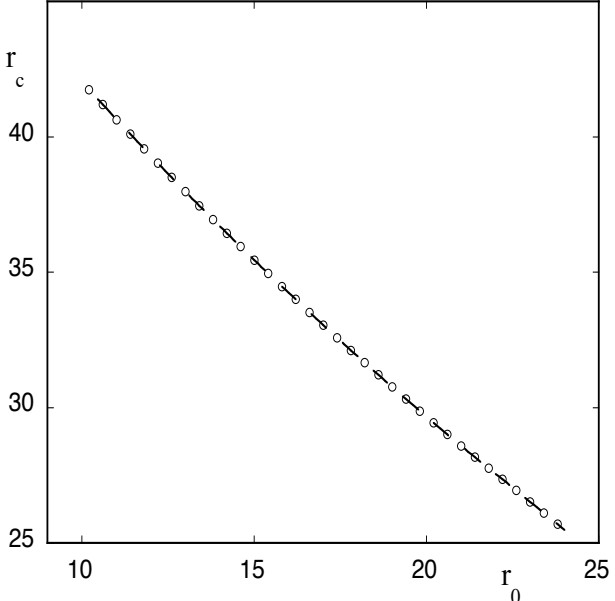

**Figure 4.**

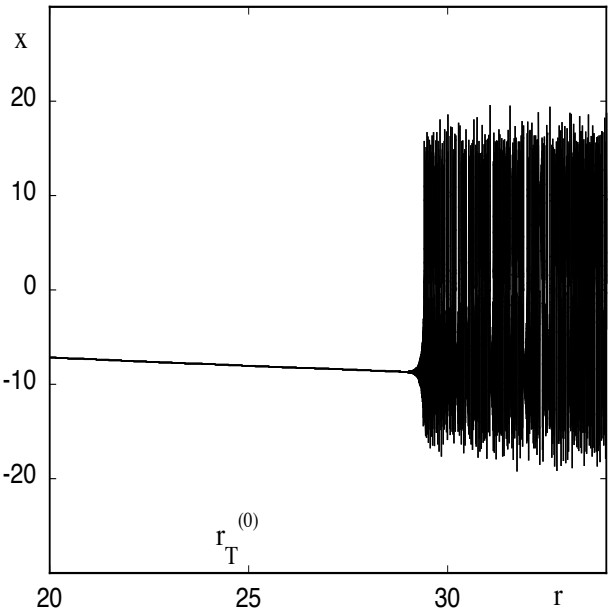

**Figure 5.**

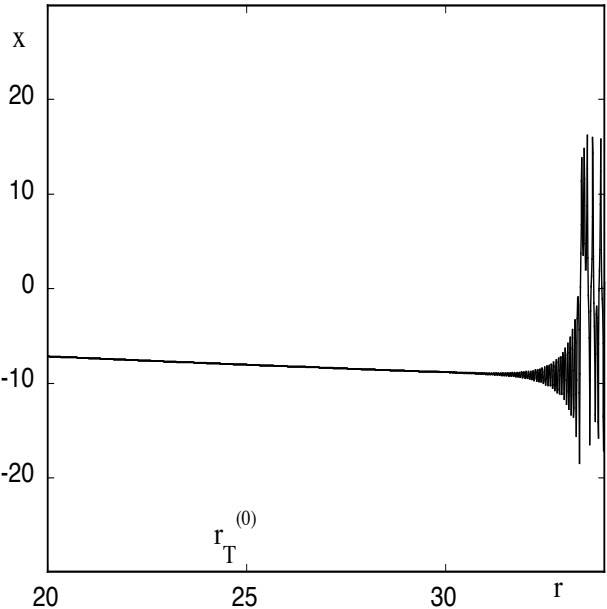

**Figure 6.**



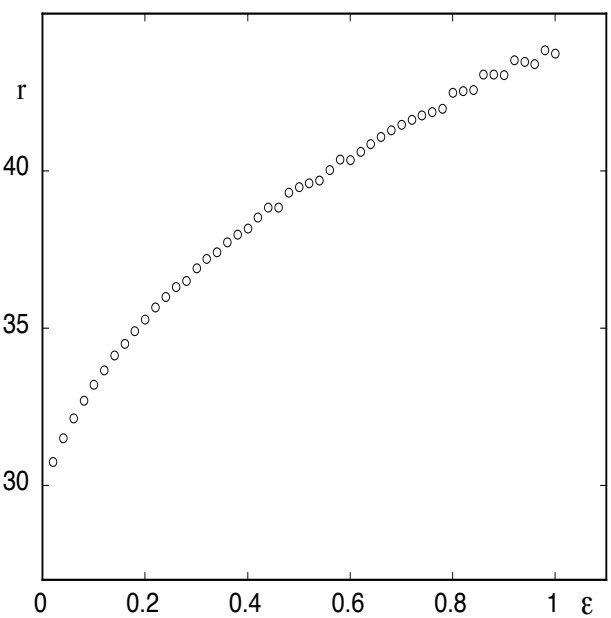

**Figure 7.**

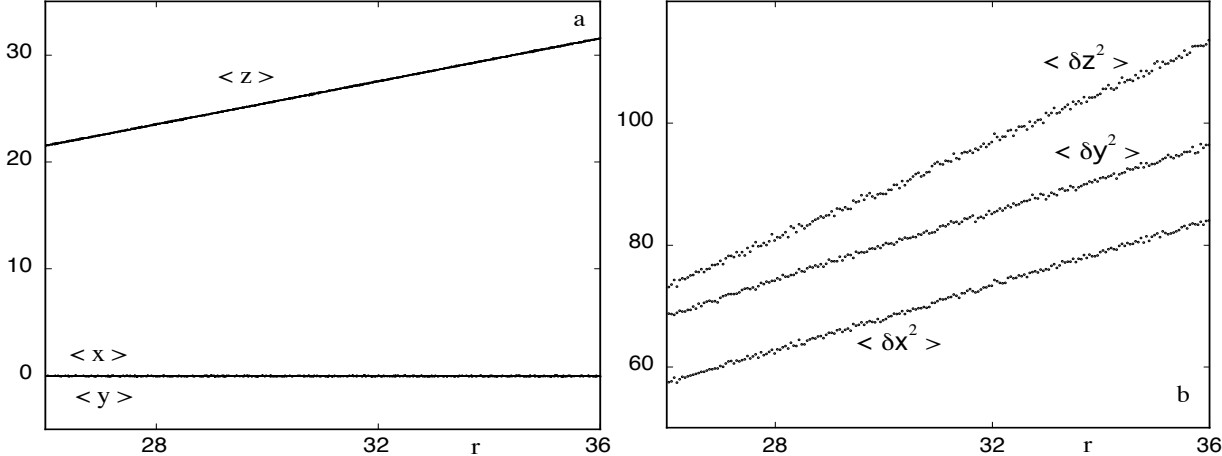

**Figure 8.**



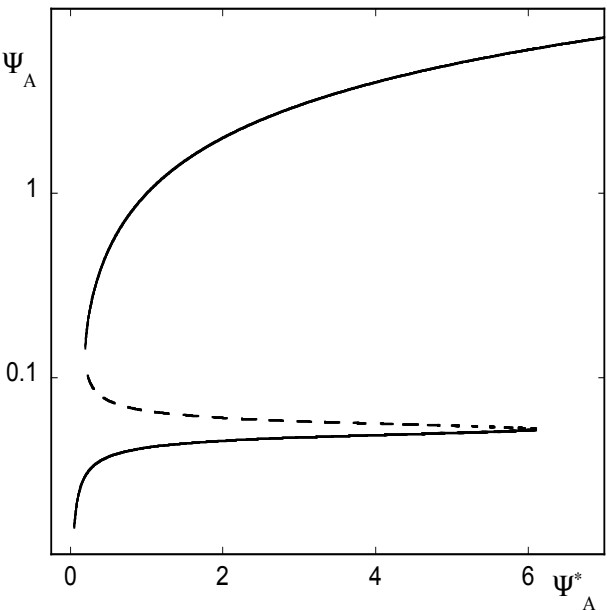

**Figure 9.**

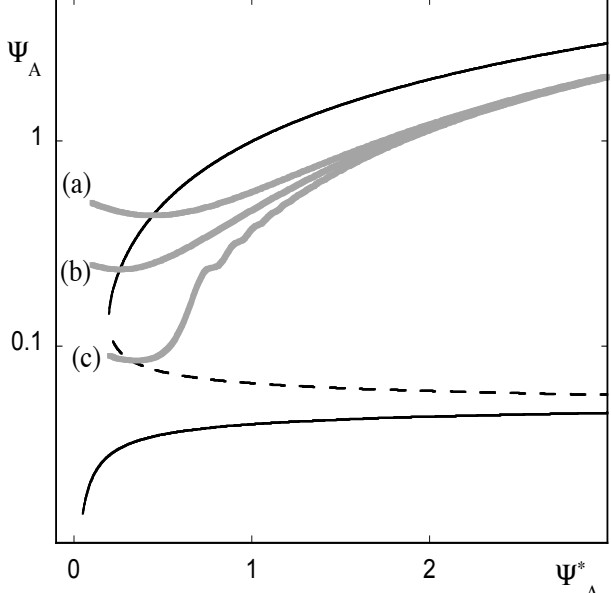

**Figure 10.**

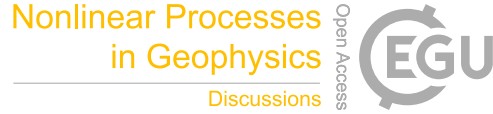

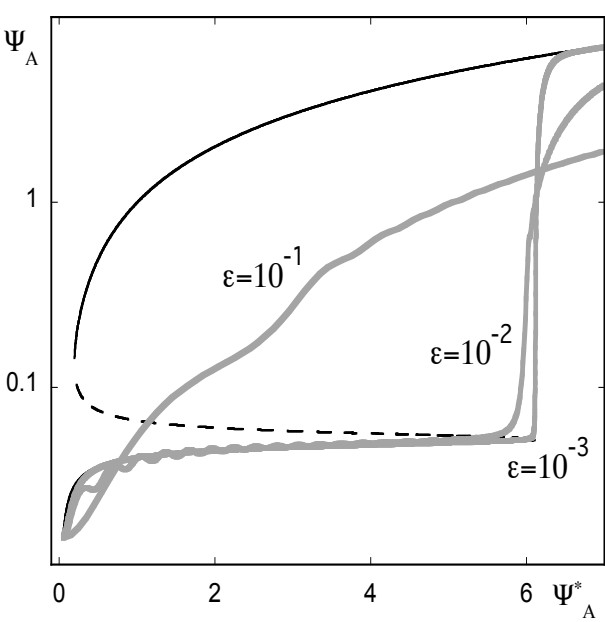

**Figure 11.**