# Peer review of "Climatic responses to systematic time variations of parameters: A dynamical approach"

_Nonlinear Processes in Geophysics, 2018_

## Referee Comment (RC1) · Anonymous Referee #1 · 5 Jul 2018

The author explores what happens to a nonlinear dynamical system (with relevance to climate) when its parameters are not just "set" but they are allowed to change as a function of time. This approach, as the author states (and I agree), is fundamentally different from the current practice in climate change projections where the parameters are changed to a new level (for example, from 1xCO2 to 2xCO2), and then the model is left to relax at a new state. It is a compelling and interesting suggestion and the results presented very illuminating. The paper is well structured and easy to follow.

---

## Referee Comment (RC2) · Anonymous Referee #2 · 11 Jul 2018

This manuscript explores the climate response to a linearly time varying parameter in general non-linear dynamical systems and in particular for a few examples of low-order models for the atmosphere-climate system. Three cases are distinguished: periodic dynamics that appear through Hopf bifurcations, chaotic dynamics using the Lorenz model for convection, and transitions between different equilibrium states which appear sometimes as back-to-back saddle node bifurcations or as limit points. The manuscript is generally well written and easy to follow. It relates to characterizing the climate response to time varying perturbations such as the atmospheric $CO_2$ concentration, which in typical climate model experiments is instantaneously doubled or quadrupled. In this study (and in reality) the $CO_2$ concentrations is however increasing in time and will lead to time dependent response in the climate system. For gen-

eral dynamical systems different types of tipping have been defined, such as purely bifurcation related tipping (corresponding to the 'static' case described here) and rate-dependent tipping (where a parameter varies in time), see Ashwin et al. 2012, http://doi.org/10.1098/rsta.2011.0306. I would suggest to relate the systems described here to these cases, at least in the discussion. Moreover, linear response theory has recently been explored to determine the transient climate response from idealized experiments where the parameter ($CO_2$) is instantaneously doubled (e.g. Lucarini 2012, http://doi.org/10.1007/s10955-012-0422-0). Would it be possible to derive from the setting described in this article conditions under which such a linear response would be valid?

---

## Author Comment (AC1) · 29 Jul 2018

I appreciate the Referee's positive assessment of my manuscript.

---

## Author Comment (AC2) · 29 Jul 2018

I appreciate the Referee's positive assessment of my manuscript and her/his constructive comments.

A revised version of the manuscript is currently being prepared in which the comments and suggestions formulated by Referee #2 are addressed. To this end the Conclusions section has been augmented and three new references have been added. Specifically:

Referee #2 Comment 1

For general dynamical systems different types of tipping have been defined, such as purely bifurcation related tipping (corresponding to the 'static' case described here) and rate-dependent tipping (where a parameter varies in time), see Ashwin et al. 2012,

http://doi.org/10.1098/rsta.2011.0306. I would suggest to relate the systems described here to these cases, at least in the discussion.

Answer

In the second paragraph of the Conclusions the following statement has been added:

"As it turned out for sufficiently small rates epsilon of parameter change a universal, epsilon-independent regime is reached in which the transition occurs at a parameter value depending entirely on the initial value and the critical value corresponding to the limit epsilon=0. But as epsilon is increased one observes rate-dependent deviations from this regime as illustrated in Figs 2, 6 and 11. Rate dependent behaviour was also reported by Ashwin et al. (2012)."

Referee #2 Comment 2

Moreover, linear response theory has recently been explored to determine the transient climate response from idealized experiments where the parameter (CO2) is instantaneously doubled (e.g. Lucarini 2012, http://doi.org/10.1007/s10955-012-0422-0). Would it be possible to derive from the setting described in this article conditions under which such a linear response would be valid?

Answer

The following paragraph has been added between the 2nd and 3rd paragraph of the Conclusions section of the original version:

"The extended stability analysis followed in this work belongs to the class of linear response theories, in the sense that it is focusing on the conditions under which perturbations, initially assumed to be small, will at some stage start to grow in time. On the other hand it is purely deterministic, as random external perturbations or intrinsic fluctuations have not been incorporated in the description. A different class of linear response theories was recently developed in climate literature (see e.g. Lucarini, 2012; Nicolis and Nicolis, 2015) in which the change in the fluctuation properties of a system

due to the presence of noise and the response of the noise free system to deterministic forcings were linked. Implicit in these approaches is the existence of a well-defined invariant probability measure of the reference system with respect to which statistical averages are carried out. Our analysis suggests that this can be so under the conditions that the system is operating around a well-defined, single stable regime, i.e. that (a) the range of variations of the forcing is nested between two successive bifurcation points; and (b), that the rate epsilon is sufficiently small so that the instantaneous perturbation to the invariant probability brought by the forcing remains small. "

---

## Author Response (AR1)

**Point by point response to the Referees and relevant changes made**

I appreciate the Referees' positive assessment of my manuscript and their constructive comments.

A revised version of the manuscript is currently being prepared in which the comments and suggestions formulated by Referee #2 are addressed. To this end the Conclusions section has been augmented and three new references have been added. Specifically:

Referee #2 Comment 1

For general dynamical systems different types of tipping have been defined, such as purely bifurcation related tipping (corresponding to the 'static' case described here) and rate-dependent tipping (where a parameter varies in time), see Ashwin et al. 2012, http://doi.org/10.1098/rsta.2011.0306. I would suggest to relate the systems described here to these cases, at least in the discussion.

Answer

In the second paragraph of the Conclusions the following statement has been added:

"As it turned out for sufficiently small rates epsilon of parameter change a universal, epsilon-independent regime is reached in which the transition occurs at a parameter value depending entirely on the initial value and the critical value corresponding to the limit epsilon=0. But as epsilon is increased one observes rate-dependent deviations from this regime as illustrated in Figs 2, 6 and 11. Rate dependent behaviour was also reported by Ashwin et al. (2012)."

Referee #2 Comment 2

Moreover, linear response theory has recently been explored to determine the transient climate response from idealized experiments where the parameter ($CO_2$) is instantaneously doubled (e.g. Lucarini 2012, http://doi.org/10.1007/s10955-012-0422-0). Would it be possible to derive from the setting described in this article conditions under which such a linear response would be valid?

Answer

The following paragraph has been added between the $2^{nd}$ and $3^{rd}$ paragraph of the Conclusions section of the original version:

[revised manuscript text omitted]